# The Relationship between Childhood Trauma Experiences and Psychotic Vulnerability in Obsessive Compulsive Disorder: An Italian Cross-Sectional Study

**DOI:** 10.3390/brainsci14020116

**Published:** 2024-01-24

**Authors:** Davide Fausto Borrelli, Laura Dell’Uva, Andrea Provettini, Luca Gambolò, Anna Di Donna, Rebecca Ottoni, Carlo Marchesi, Matteo Tonna

**Affiliations:** 1Department of Medicine and Surgery, Psychiatry Unit, University of Parma, 43126 Parma, Italy; davidefausto.borrelli@unipr.it (D.F.B.);; 2Department of Mental Health, Local Health Service, 29121 Piacenza, Italy; 3Department of Mental Health, Local Health Service, 43125 Parma, Italy

**Keywords:** OCD, psychosis, schizophrenia, vulnerability, trauma, compulsions

## Abstract

People with obsessive compulsive disorder (OCD) are at increased risk of developing psychotic disorders; yet little is known about specific clinical features which might hint at this vulnerability. The present study was aimed at elucidating the pathophysiological mechanism linking OCD to psychosis through the investigation of childhood trauma experiences in adolescents and adults with OCD. One hundred outpatients, aged between 12 and 65 years old, were administered the Yale–Brown Obsessive Compulsive Scale (Y-BOCS) and its Child version (CY-BOCS), as well as the Childhood Trauma Questionnaire (CTQ); Cognitive–Perceptual basic symptoms (COPER) and high-risk criterion Cognitive Disturbances (COGDIS) were assessed in the study sample. Greater childhood trauma experiences were found to predict psychotic vulnerability (*p* = 0.018), as well as more severe OCD symptoms (*p* = 0.010) and an earlier age of OCD onset (*p* = 0.050). Participants with psychotic vulnerability reported higher scores on childhood trauma experiences (*p* = 0.02), specifically in the emotional neglect domain (*p* = 0.01). In turn, emotional neglect and psychotic vulnerability were found higher in the pediatric group than in the adult group (*p* = 0.01). Our findings suggest that childhood trauma in people with OCD may represent an indicator of psychotic vulnerability, especially in those with an earlier OCD onset. Research on the pathogenic pathways linking trauma, OCD, and psychosis is needed.

## 1. Introduction

In the last decade, there has been a renewed interest in the complex interplay between obsessive compulsive disorder (OCD) and schizophrenia spectrum disorders (SSD) due to certain intertwining features in clinical presentation and course of illness between the two disorders. In fact, OCD and SSD are characterized by apparently overlapping clinical characteristics (i.e., sense of intrusive and recurrent unpleasant thoughts), which are actually underpinned by different phenomenological experiences. In fact, the experiential “mineness” of intrusive thoughts (i.e., whether they are perceived as belonging to the self or not) [1,2] is preserved in people with OCD, while it is questioned in people with SSDs due to a weakening of the primitive feeling that one’s own thoughts are self-generated [3,4]. Furthermore, the two disorders present high rates of comorbidity, with prevalence rates around 12% [5], and are overlapping in terms of age of onset and course of illness [6]. Longitudinal studies showed that a primary diagnosis of OCD tends to increase the subsequent risk of developing SSD [7,8,9], while cross-sectional studies highlighted the possibility that pediatric forms of OCD may represent a red flag for an underlying schizophrenia vulnerability [10,11,12]. In a similar vein, the genetic [13,14], neuroimaging [15,16], and neurocognitive [17] literature indicate the possibility of overlapping neurocircuitries in OCD and SSDs. However, putative biomarkers of psychotic vulnerability in the OCD population have been scarcely investigated yet; the same applies to the underlying pathophysiological mechanisms beneath their longitudinal comorbidity. This may be because most studies have focused on the role of OCD in patients with SSD (for a comprehensive summary, see Cunill et al., 2023) [18], thus neglecting the study of psychotic vulnerability in people with OCD. In particular, inadequate attention has been accorded to specific clinical features of childhood OCD that in fact could represent putative markers of psychotic progression in adulthood. 

In this direction, the investigation of traumatic experiences might be a promising field of research to capture the link between the two disorders. Trauma exposure can produce a complex array of long-lasting physiological, emotional, and cognitive reactions that might cause stable phenotypic changes [19], which are predictive of later psychopathology in adults [20]. On the one hand, childhood trauma experiences are associated with threefold increased odds of subsequent first onset of psychotic symptoms [21] and can predict the psychotic onset in “ultra-high risk” patients [22,23]. On the other, there is evidence for an important role of trauma in the development of OCD. In fact, 50–70% of people with OCD report trauma experiences before the onset of obsessive compulsive symptoms (OCS) [24]. In this connection, cognitive–behavioral approaches have suggested that adverse early experiences may lead to the development of cognitive schemas related to difficulty with unpredictability, novelty, and change, with an excessive need for control [25,26], all predisposing conditions for the development of OCD [27,28]. As to the possible relationship between trauma events and clinical features of OCD, while some studies failed to find a significant association [29,30], others have highlighted a relationship between OCS severity [31,32,33] and an earlier age of OCD onset [34,35]. With specific regard to the pediatric population, a possible link between OCD and trauma is even less clear. Vazequez et al. 2022 [36] have found that a history of trauma would not exert a significant impact on the severity of the disorder, whereas a recent systematic review [37] showed mixed findings on this topic_,_ arguing for more knowledge on this age group.

We suggest that a dimensional perspective might be useful to better understand the association between trauma and OCD, while also attempting to elucidate the pathophysiological mechanism linking OCD to psychosis. In this vein, in a large sample of nonclinical youth, it was found that participants exposed to trauma showed higher rates of psychosis if they also presented OCS, suggesting that young people with prominent OCS and a documented history of trauma may be suspected of having an underlying psychotic vulnerability [38]. However, to date, these findings have not been replicated on a clinical sample. 

Therefore, the main aim of our research was to examine the connection between childhood trauma experiences and psychotic vulnerability in a sample of adolescents and adults with OCD. We expected a greater history of trauma to be associated with a greater likelihood of psychotic vulnerability, a more severe presentation of OCD, and an earlier OCD onset. Moreover, after splitting the study sample according to psychotic vulnerability and age, we expected higher scores of childhood trauma in OCD patients with psychotic vulnerability and no differences in past trauma experiences between the pediatric and the adult group.

## 2. Materials and Methods

### 2.1. Participants

The present study enrolled a sample of 100 OCD outpatients aged between 12 and 65 years old. Pediatric participants were recruited from January 2021 to January 2023 from the Neuropsychiatric Clinic of Parma’s Childhood and Adolescence Service (Parma, Italy). Adult participants (18–65 y.o.) were recruited from the Psychiatric Unit of the University of Parma (Parma, Italy) between May 2018 and January 2023. Written informed consent was provided by each participant (parents completed the consent form for patients under the age of 18). Criteria for inclusion were the following: (1) OCD diagnosis based on DSM-5 criteria (APA, 2013) [39] and (2) a signed informed consent form to be included in the study. The following conditions were considered as exclusion criteria: (1) a mental disorder that was currently associated with a general medical condition; (2) affective or psychotic comorbidity to the OCD diagnosis; (3) a condition that was currently associated with drug or alcohol abuse or dependence; and (4) a cognitive disorder (Mini-Mental State Examination score < 25) that could potentially interfere with testing procedures.

All adult participants were in treatment with SSRIs; among them, nonresponders (17.3%) were also treated with antipsychotics. In the pediatric sample, there was a mix of patients receiving therapy (*n* = 20) and patients who were referred by physicians for a preliminary evaluation (*n* = 28). Of those receiving treatment, 20.8% received SSRIs, 14.6% received SSRIs enhanced with antipsychotics, and 4.2% received SSRIs in addition to psychotherapy.

After ruling out any conditions that would impair their understanding of the protocol questions, every patient received an exhaustive and comprehensive explanation of the study. The study was approved by the Local Institutional Ethics Committee of “Area Vasta Emilia Nord” (Emilia-Romagna region, approved on 6 October 2020, n.35564). 

### 2.2. Clinical Assessment

Clinical evaluation was conducted in two sessions: in the first, sociodemographic data (i.e., gender, age, age at onset) were collected; SCID-5 CV [40] was conducted to confirm the diagnosis of OCD, study questionnaires were administered, and OCS severity was assessed. In the second, performed on average one week apart, psychotic vulnerability was assessed, and questionnaires were collected. A trained psychiatrist (D.F.B.) completed each assessment.

The evaluations were carried out under the supervision of the participants’ clinicians. During the assessment, pediatric participants could have their parents/guardians participate if requested. All the participants were offered a moment of restitution for the tests performed, during which clinicians and, in the case of pediatric participants, parents/guardians were involved. Similarly, each patient was given the opportunity to report on their experience with the evaluations carried out. Each participant was given the possibility to retire from the study at any time.

### 2.3. Measures

#### 2.3.1. Yale–Brown Obsessive Compulsive Scale (Y-BOCS) and Children’s Yale–Brown Obsessive Compulsive Scale (CY-BOCS)

In the adult sample, OCD severity was evaluated using the Yale–Brown Obsessive Compulsive Scale (Y-BOCS) [41], while in the pediatric sample, it was determined using the Children’s Yale–Brown Obsessive Compulsive Scale (C-YBOCS) [42]. Both scales are semi-structured interviews that do not depend on specific types of symptoms (e.g., washing, control) but on the level of severity of symptoms reported by the interviewee. On a scale of 0 (no symptoms) to 4 (severe symptoms), each symptom-related item in the week prior to the interview is rated: (1) time spent; (2) interference degree; (3) distress; (4) resistance; and (5) felt control over the condition. Recently, OCD severity cut-offs have been empirically defined as follows: 0–13 scores defined subclinical OCD; 14–21 scores mild OCD; 22–29 scores moderate OCD; and 30–40 scores severe OCD [43]. The interview took approximately 60 min to administer.

#### 2.3.2. Childhood Trauma Questionnaire (CTQ)

The Childhood Trauma Questionnaire (CTQ) [44] was utilized to determine the occurrence of traumatic experiences in childhood. CTQ is a self-report questionnaire widely used to measure the severity of five different forms of childhood trauma (i.e., emotional abuse, emotional neglect, physical abuse, physical neglect, and sexual abuse) and participants’ tendency to underreport maltreatment.

#### 2.3.3. Schizophrenia Proneness Instrument—Adult version (SPI-A) and Schizophrenia Proneness Instrument—Child and Youth version (SPI-CY)

Psychotic vulnerability was assessed using COPER and COGDIS criteria from the Schizophrenia Proneness Instrument—Adult version (SPI-A) [45] and the Schizophrenia Proneness Instrument—Child and Youth version (SPI-CY) [46]. The interviews were administered by D.F.B., who completed an approved training in these instruments. Each COPER and COGDIS item was rated on a 7-point severity scale according to the maximum frequency in the last 3 months. Only subjective experiences that are currently present are considered in the interview. Hence, an experience cannot be considered as present by the interviewer unless it is described in depth by the individual. The COPER criteria are satisfied when at least one of the symptoms (see Table 1) has occurred at least once a week over the past three months and the symptoms originally appeared more than a year before the examination. The COGDIS criteria are met when at least two of the symptoms (see Table 1) have occurred at least once a week for the last three months. Moreover, symptoms need not have been associated with drug use or be present during the premorbid stage of the illness. For each patient in our investigation, we indicated whether they met the COPER and/or COGDIS criteria. The SPI-A and SPI-CY took approximately 90 min to administer.

#### 2.3.4. Social and Occupational Functioning Assessment Scale (SOFAS)

Using the Social and Occupational Functioning Assessment Scale (SOFAS) [47], social occupational functioning has been evaluated in both adults and children. 

### 2.4. Statistics

R Studio (version 2021.09.0) was used for all statistical analyses. In order to analyze the clinical and demographic factors across the entire sample, descriptive statistics were first used.

Second, we estimated the correlation coefficient for every pair of variables to investigate relationships between the clinical and demographic characteristics. The total sample size (*n* = 100) exceeded the minimum amount required (*n* = 67) as estimated by means of a statistical a priori sample size calculation obtained with the bivariate normal model (1 – β = 0.8, α = 0.05). Due to the ordinal nature of some of the variables, we used the R library *latentcor*, which uses the latent Gaussian copula model to properly estimate correlation coefficients among various variable types (e.g., ordinal and zero-inflated variables). 

Third, using R library *caret*, a logistic regression model was performed to assess whether psychotic vulnerability could be explained by trauma experiences, OCD severity, and age of OCD onset. We run a multiple logistic regression to predict the probability of class membership (i.e., psychotic vulnerability) based on multiple predictor variables (i.e., trauma experiences, OCD severity, and age of OCD onset). We used the *pR2* function from the *pscl* package to estimate McFadden’s R^2^ for our model. Therefore, we evaluated the importance of each predictor variable in the model by using the *varImp* function from the *caret* package. Last, we used the *plotROC* package to represent the receiver operating characteristics (ROC) curve of our model and to estimate its area under the curve (AUC).

Finally, we compared patients who met psychotic vulnerability criteria (COPER and/or COGDIS criteria) with patients without psychotic vulnerability. Similarly, we compared the pediatric and adult groups on their trauma experiences (CTQ total score and CTQ domains scores). Independent sample *t*-tests were used for group comparisons. In both cases, the total sample size was similar to the minimum amount required (*n* = 102), estimated by means of a statistical a priori sample size calculation obtained by an independent *t*-test (1 − β = 0.8, α = 0.05, and effect size d = 0.5)

## 3. Results

### 3.1. Participants

One hundred OCD outpatients (mean age = 29.20 ± 17.01) were recruited in the present study, including 56 females (56%) and 44 males (44%). Table 2 reports the sociodemographic and clinical characteristics of the study sample. Overall, 32 patients (32% of the sample) met COPER (32% of the sample) or COGDIS criteria (28% of the sample) (i.e., being classified as having psychotic vulnerability). The pediatric group included 48 patients (48%, mean age = 15.62), and the adult group had 52 patients (52%, mean age = 42.78). 

### 3.2. Correlations among Psychopathological Variables in the Whole Sample 

Psychotic vulnerability was positively associated with the Y-BOCS total score (r = 0.38, *p* < 0.001), Y-BOCS obsessions (r = 0.27, *p* < 0.01), Y-BOCS compulsions (r = 0.26, *p* < 0.01), CTQ total score (r = 0.39, *p* < 0.001), and CTQ emotional neglect score (r = 0.37, *p* < 0.001), while it was negatively correlated with age of OCD onset (r = −0.40, *p* < 0.001). The patterns of correlation among childhood trauma experiences and demographic and psychopathological variables are reported in Table 3.

### 3.3. Logistic Regression 

A logistic regression was performed to establish the effects of age of onset, experiences of trauma (CTQ total score), and OCD severity (YBOCS total score) on the likelihood that OCD participants met COPER/COGDIS criteria. The McFadden’s R^2^ of our sample was 0.14. According to McFadden, values from 0.2 to 0.4 indicate excellent model fit [48]. Increasing CTQ total score (coefficient estimate = 0.052, *p* = 0.018), YBOCS total score (coefficient estimate = 0.087, *p* = 0.010), and decreasing age of OCD onset (coefficient estimate = −0.052, *p* = 0.050) were associated with an increased likelihood of psychotic vulnerability (COPER and/or COGDIS criteria). Therefore, we computed the importance of each predictor variable in the model: YBOCS total score was the most important predictor variable (2.57), followed by CTQ total score (2.35) and then the age of OCD onset (1.89). Thus, we used a receiver operating characteristics (ROC) curve to represent the performance of our model (see Figure 1). The area under the curve (AUC) of the ROC curve was 0.77. 

### 3.4. Comparison between Groups 

#### 3.4.1. Patients with Psychotic Vulnerability vs. Patients without Psychotic Vulnerability

Compared to patients without psychotic vulnerability, the group with psychotic vulnerability reported higher scores on the CTQ total score (t[−9.99; −0.87] = −2.36, *p* = 0.02) and on the emotional neglect subscale (t[−4.09; −0.60] = −2.66, *p* = 0.01) (see Table 4). 

#### 3.4.2. Pediatric Group vs. Adult Group

The pediatric group reported higher scores on the emotional neglect (t[0.49; 3.75] = 2.58, *p* = 0.01) subscale than the adult group. The differences between the groups are reported in Table 4. Therefore, there was a significant association between being part of the pediatric and meeting COPER/COGDIS criteria (χ^2^ (1) = 3.96, *p* = 0.038).

## 4. Discussion

The present study was aimed at investigating the patterns of association between childhood trauma experiences and psychotic vulnerability in a sample of adolescents and adults with OCD.

First, we found that psychotic vulnerability (i.e., meeting COPER/COGDIS criteria) was positively associated with more severe OCD (YBOCS total score) and greater childhood trauma experiences (CTQ total score), in particular in the emotional neglect domain (CTQ Emotional Neglect score), whereas it was negatively correlated with the age of OCD onset. The results were confirmed by the regression analysis model, in which psychotic vulnerability was predicted by increasing scores in OCD severity and childhood trauma experiences, as well as an earlier age of OCD onset. The latter is consistent with the strong evidence of a relationship between an earlier age of OCD onset and psychosis proneness in people with OCD [11,12]. Instead, the mechanisms underlying the complex interplay between psychotic vulnerability, OCD severity, and childhood trauma experiences in subjects with OCD remain speculative. In a recent study of our group [49], we found that in people with OCD, childhood trauma severity was associated with higher OCD severity and, particularly, with a more complex structure of OCD compulsions. In this regard, we argued that, in predisposing individuals, different interpersonal traumatic events might lead to compulsive behaviors, as an attempt to manage intrusive trauma-related negative feelings. Similarly, in another study from our group [50], we found that a more complex structure of OCD compulsions was also associated with early signs of psychosis. Therefore, it is intriguing to hypothesize that, in some cases, the onset of OCD symptomatology, particularly at an earlier age, might reflect a developmental mechanism to cope with different sources of underlying biopsychosocial unpredictability [51,52]. Moreover, we cannot exclude that behavioral concomitant trauma, such as avoidance and withdrawal, may limit opportunities to seek adaptative feedback from everyday experiences, especially for young people with severe OCD, allowing psychotic explanations to develop.

Second, after collapsing the total sample into two groups according to the presence of COPER and/or COGDIS criteria, we found that OCD participants who met COPER/COGDIS criteria (i.e., with psychotic vulnerability) reported higher scores on childhood trauma experiences (CTQ total score), in particular in the emotional neglect domain (CTQ Emotional Neglect). The latter is in line with the recent literature, suggesting a specific role of childhood neglect in psychosis [53]. We speculate that a history of childhood emotional neglect, more than other traumatic experiences, may lead to an impairment of social cognition processes, as observed in SSDs [54]. In fact, childhood emotional neglect has been found to be associated with impaired recognition of familiar stimuli [55] and a slower ability to identify the valence of emotional faces [56]. On the contrary, in studies on trauma-related OCD, all interpersonal traumatic events, interpreted as unpredictable and uncontrollable, were found to be prevalent in the years preceding an OCD diagnosis [57,58]. Overall, according to the results from Barzilay and colleagues’ study in a nonclinical population [38], our data would confirm that an underlying psychotic vulnerability should be investigated in youths with prominent OCS and a reported history of trauma. In a similar vein, our findings are consistent with the evidence that psychotic patients with OCS reported more childhood traumatic events compared to those without OCS [59]. Therefore, we speculate that the co-occurrence of childhood trauma experiences and prominent OCS should be recognized as nonspecific risk factors of underlying psychotic vulnerability, especially in developmental years.

Third, after dividing the study sample based on age, we found no significant differences between the pediatric and adult groups in terms of overall past trauma experiences (CTQ total score). This finding seems to rule out a hypothetical tendency to overestimate trauma experiences in adolescents with OCD, as well as a tendency to underestimate previous trauma experiences in adults with OCD. Therefore, self-report measures of childhood trauma appear stable over time in people with OCD. However, we found significantly higher rates of emotional neglect and higher psychotic vulnerability (i.e., COPER and/or COGDIS symptoms) in the pediatric group than in the adult group. This result further confirms the close relationship between childhood neglect and the development of psychotic symptoms in early-onset OCD [60,61], worthy of longitudinal investigation.

Overall, the findings of the present study would confirm an association between childhood trauma experiences and psychotic vulnerability in people with OCD. These findings may also be interpreted in the light of recent neurobiological evidence. In particular, the relationship between trauma and psychosis in OCD patients might be underpinned by a dysregulation of the immune system. In fact, on the one hand, a large body of literature supports the hypothesis of the involvement of the immune system in the pathophysiology of OCD, especially in childhood [62,63]. On the other, there is increasing evidence for a relationship between chronic proinflammatory states and schizophrenia [64]. Moreover, childhood traumatic experiences seem to play a role in the dysregulation of peripheral immune activity of people with schizophrenia [65]. Therefore, it is intriguing to hypothesize a possible mediating role of trauma experiences on the immune system in patients with early OCD onset and psychotic vulnerability.

Some methodological limitations deserve to be addressed. First, the cross-sectional design of the study should be taken into account, as it cannot exclude that obsessive compulsive and pre-psychotic symptoms may change over time or have a phase-dependent effect. In the same vein, we cannot draw firm conclusions about the etiological influence of childhood trauma experiences over OCD course and phenomenology. Second, our results should be considered with the caveat of the small sample size. Larger prospective studies are needed to confirm our results, especially to assess the accuracy (i.e., the proportion of correct predictions over total predictions) of our logistic regression model. Future prospective studies, also focusing on the interplay between mediating and moderating factors (e.g., environmental factors), should be addressed to unravel the developmental trajectories linking childhood trauma experiences, psychotic vulnerability, and OCD onset. Third, the age of onset was determined retrospectively, based on reports from clinicians and family members; therefore, this method is sensitive to recall bias. Moreover, the inclusion of both pediatric and adult participants in our study sample may be questionable, as childhood OCD might actually represent a distinct subtype of OCD with different comorbidity patterns, clinical features, and course compared to adult OCD [66]. Therefore, our results should be considered preliminary. Future research on larger samples of pediatric participants is needed to more precisely define the impact of age of onset on the clinical presentation and developmental trajectories of OCD. Fourth, there results might be biased by the major diagnostic instability of OCD diagnosis in pediatric patients with respect to adults [66]. Fifth, diagnostic comorbidities were not considered (e.g., anxiety, tic, and affective disorders), nor was the duration of untreated illness. Sixth, the investigation of traumatic experiences was limited to the domains of the CTQ, which is mainly based on the examination of the family environment. For this reason, we cannot exclude that other types of traumatic experiences may be relevant, such as early childhood economic deprivation, bullying, or exposure to traumatic stressors (e.g., natural disasters or accidents). Seventh, the psychiatric family history was not investigated. It is therefore not possible to exclude that individuals who have met the COPER/COGDIS criteria may have a positive family history of SSDs. Eighth, the potential confounding effect of psychopharmacological treatments and psychotherapy on OCD severity was not taken into account. Lastly, correlation analysis was not corrected for multiple comparisons.

## 5. Conclusions

Despite the above limitations, our findings suggest an association between past trauma experiences, psychotic vulnerability, and early-onset OCD. In this connection, OCS might represent a “homeostatic” mechanism to give order and predictability over the developmental trajectories linking trauma and psychosis. These findings pave the way for future research on the pathogenic pathways linking trauma, OCD, and schizophrenia. In this direction, attention should be focused on targeted interventions to empower effective coping strategies in young individuals with trauma-related OCD.

## Figures and Tables

**Figure 1 brainsci-14-00116-f001:**
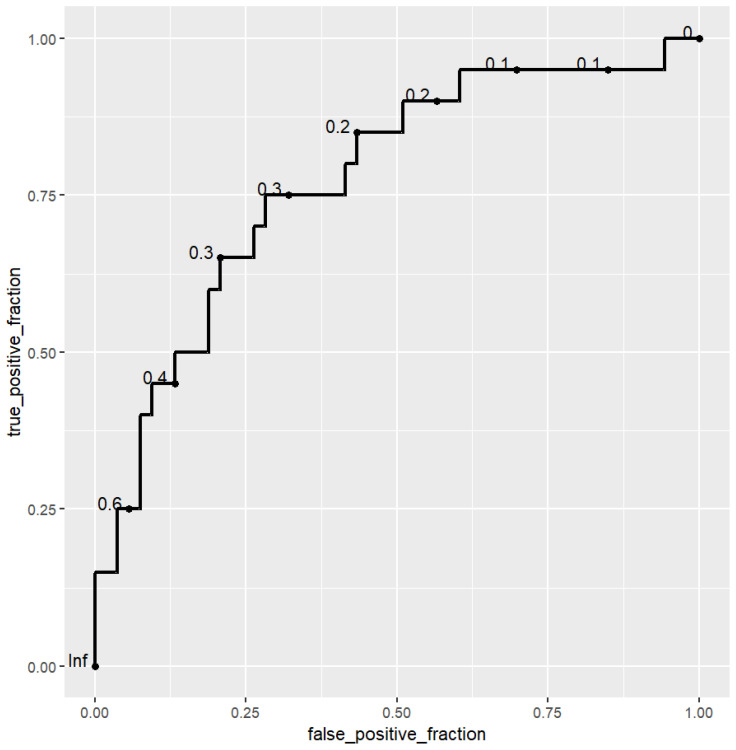
Receiver operating characteristics (ROC) curve of our model.

**Table 1 brainsci-14-00116-t001:** Cognitive–Perceptual basic symptoms (COPER) and high-risk criterion Cognitive Disturbances (COGDIS).

**Risk Criterion/Cognitive–Perceptive basic symptoms (COPER)** COPER criteria are met when at least one of the following symptoms is present with a score of ≥3, with a minimum weekly occurrence during the last three months, and the onset of symptoms (or the worsening of a pre-existing symptom) occurred more than a year before the evaluation. Only subjective experiences that are currently present are considered in the interview. A symptom cannot be considered as present by the interviewer unless it is described in depth by the individual.
Thought interference experiences;
Thought perseveration experiences;
Thought pressure experiences;
Thought blockages experiences (≥13 y.o.);
Experiences of receptive language disturbances (visual and/or spoken);
A lesser ability to discern between ideas and perceptions, fantasies, and true memories;
The occurrence of unstable ideas of reference;
Derealization;
Experiences of visual perception aberrations, including partial vision;
Experiences of auditory perception aberrations, including tinnitus.
**High-Risk Criterion/Cognitive Disturbances basic symptoms (CODGIS)** COGDIS criteria are met when at least two of the following symptoms, assessed over the last three months, have a score of ≥3. Only subjective experiences that are currently present are considered in the interview. As for COPER criteria, a symptom cannot be considered as present by the interviewer unless it is described in depth by the individual.
A lesser ability to divide attention;
Interference experiences of emotionally neutral thoughts;
Thought pressure experiences;
Thought blockages experiences (≥13 y.o.);
Experiences of receptive language disturbances (visual and/or spoken);
Disturbances of expressive language;
The occurrence of unstable ideas of reference;
Disturbances of abstract thinking, concrete thinking (≥13 y.o.);
Visual details capture the attention of the subject.

Note. Description of COPER and CODGIS symptoms as per Schizophrenia Proneness Instrument—Adult version [45] and Schizophrenia Proneness Instrument—Child and Youth version [46].

**Table 2 brainsci-14-00116-t002:** Sociodemographic and clinical features of the study sample.

	Patients (*n* = 100)	
	*n*	
Gender		
Male	44	
Female	56	
Group		
Pediatric Group	48	
Adult Group	52	
Age of onset		
Before age of majority	59	
After age of majority	41	
Psychotic vulnerability		
No psychotic vulnerability	68	
With psychotic vulnerability	32	
		Mean (SD)
Age (years)		29.20 (±17.01)
Age of onset (years)		18.87 (±9.93)
Years of illness		10.33 (±11.28)
Psychopathological variables		
Y-BOCS/CY-BOCS total		21.34 (±7.78)
Y-BOCS/CY-BOCS obsessions		10.86 (±4.47)
Y-BOCS/CY-BOCS compulsions		10.44 (±4.41)
CTQ total score		38.71 (±10.96)
CTQ Emotional Neglect		12.25 (±4.22)
CTQ Emotional Abuse		7.65 (±3.40)
CTQ Physical abuse		6.38 (±2.95)
CTQ Physical Neglect		6.34 (±2.35)
CTQ Sexual abuse		5.88 (±2.91)
SOFAS		65.44 (±12.45)

Note. Y-BOCS = Yale–Brown Obsessive Compulsive Scale; CY-BOCS = Children’s Yale–Brown Obsessive Compulsive Scale; CTQ = Childhood Trauma Questionnaire; SOFAS = Social and Occupational Functioning Assessment Scale.

**Table 3 brainsci-14-00116-t003:** Heatmap between childhood trauma experiences and demographic and psychopathological variables.

	CTQ Total	CTQ EN	CTQ EA	CTQ PA	CTQ PN	CTQ SA
Gender	0.03	0.05	0.02	−0.03	−0.03	0.21 *
Age	−0.02	−0.08	−0.16	−0.01	0.29 **	0.13
Age of onset	−0.04	−0.05	−0.18	0.06	0.21 *	0.18
Years of illness	−0.01	−0.09	−0.1	−0.09	0.22 *	0.01
Y-BOCS total	0.11	0.03	0.16	0.1	−0.01	0.06
Y-BOCS obsessions	0.08	0.08	0.12	0.06	−0.02	0.04
Y-BOCS compulsions	0.15	0.03	0.16	0.15	0.05	−0.01
Psychotic vulnerability	0.29 **	0.21 *	0.28 **	0.11	0.15	0.12

Note. Colored cells are used to represent the correlation coefficient magnitude and the positivity/negativity of the correlation. The colors of the spots are used to depict the direction of the correlation (blue for negative; red for positive) and the dimension of the correlation magnitude (the stronger the color, the larger the correlation coefficient). The significant associations are indicated with stars (* *p* < 0.05; ** *p* < 0.01). CTQ total = Childhood Trauma Questionnaire total score; CTQ EN = Childhood Trauma Questionnaire Emotional Neglect domain; CTQ EA = Childhood Trauma Questionnaire Emotional Abuse domain; CTQ PA = Childhood Trauma Questionnaire Physical Abuse domain; CTQ PN = Childhood Trauma Questionnaire Physical domain; CTQ SA = Childhood Trauma Questionnaire Sexual Abuse domain.

**Table 4 brainsci-14-00116-t004:** Comparison between groups.

	Patients with Psychotic Vulnerability	Patients without Psychotic Vulnerability			
*N*	32	68			
Gender (female)	20	36		F^+^ = 0.81 *p* > 0.05	
	Mean (SD)	Mean (SD)	t	*p*	d
CTQ total score	42.40 (11.37)	36.97 (10.41)	−2.365	0.020 *	0.5
CTQ Emotional Abuse	8.13 (3.28)	7.43 (3.46)	−0.958	0.34	0.21
CTQ Emotional Neglect	13.84 (4.13)	11.50 (4.07)	−2.668	0.009 **	0.57
CTQ Physical Abuse	6.47 (2.60)	6.34 (3.1)	−0.205	0.827	0.04
CTQ Physical Neglect	7.09 (3.32)	5.98 (1.63)	−2.241	0.082	0.48
CTQ Sexual Abuse	6.62 (3.97)	5.53 (2.19)	−1.775	0.079	0.38
	Pediatric Group	Adult Group			
*N*	48	52			
Gender (female)	28	28		F^+^ = 0.82 *p* > 0.05	
	Mean (SD)	Mean (SD)	t	*p*	d
CTQ total score	39.27 (9.36)	38.19 (12.34)	0.495	0.622	0.09
CTQ Emotional Abuse	8.06 (3.34)	7.27 (3.44)	1.169	0.245	0.23
CTQ Emotional Neglect	13.35 (3.88)	11.23 (4.30)	2.595	0.011 *	0.51
CTQ Physical Abuse	6.56 (3.13)	6.21 (2.79)	0.589	0.557	0.11
CTQ Physical Neglect	5.92 (1.50)	6.73 (2.89)	−1.746	0.084	−0.34
CTQ Sexual Abuse	5.37 (1.90)	6.35 (3.56)	−1.68	0.096	−0.33

Note. ** *p* ≤ 0.01; * *p* ≤ 0.05.

## Data Availability

The data that support the findings of this study are available from the corresponding author, D.F.B., upon reasonable request. The data are not publicly available because the participants of this study did not give written consent for their data to be shared publicly.

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
