# Peer review of "The Relationship between Childhood Trauma Experiences and Psychotic Vulnerability in Obsessive Compulsive Disorder: An Italian Cross-Sectional Study"

_brainsci, 2024, doi:10.3390/brainsci14020116_

Round 1
Reviewer 1 Report
Comments and Suggestions for Authors
This study is of interest following a clear design and presents new results.
In principle the paper can be accepted as it is. Nevertheless there is a surprising gap in the overall outline, in that exclusively cognitive neurodevelopmental pathogenic perspective is discussed (both in introduction and in discussion of the study). However there is in the meanwhile broad evidence that childhood trauma can induce immunological abnormalities and that immunological abnormalities are involved in both OCD and psychosis., potentially sugggesting a different or at least alternative pathophysiological link as compared to the discussed one. Ithink this should be included....
Minor points: there are 2 spelling errors :
in table 1 ...ideas of reference(not refence)
in table 3: " red" is missing I think in context with "for positive"
Reviewer 2 Report
Comments and Suggestions for Authors
The manuscript is well-written by and large and the topic is a kind of important. The manuscript can be improved considering following comments:
1. Place and type of study should be added to the title.
2. Main statistical data and p-value should be added to Abstract.
3. Name of city and country of study should be added to Methods.
4. Ethical consideration, particularly facing patient with severe illness or a children with ongoing abuse, should be explained clearly.
5. In my opinion, percentage column in Table-2 is not useful.
6. Instruments should be described separately in details.
7. What`s red in Table-3? Caption should be edited carefully.
8. Biological aspect of the findings and topic should be discussed briefly as well.
Reviewer 3 Report
Comments and Suggestions for Authors
This paper examines the association between adverse childhood experiences and proneness to psychosis in patients with obsessive-compulsive disorder (OCD).
There is an extensive literature on the links between OCD and the psychotic spectrum, and the research reported in this paper is a potentially valuable addition to this literature as it examines the effects of childhood adversity / trauma in relation to the relationship between the two. The authors have used standard, well-validated instruments to assess all study variables, and I did not detect any major methodological errors in the work as presented. The authors have also frankly acknowledged the limitations of their work and have presented their results in a balanced and accurate manner.
The following are aspects of the paper that would benefit from correction or clarification, as appropriate:
1. It is preferable to use uniform, standard terminology when referring to outcome variables. In the paper (title, abstract, text), both "psychotic disposition" (a more general and somewhat vague term) and "psychotic vulnerability" (a more precise term indicating a high-risk state) are used interchangeably. I would suggest using the latter term throughout.
2. The introduction should provide a clearer description of the links between OCD and schizophrenia spectrum disorders (SSD). The link mentioned by the authors ("thoughts which are difficult to control") is phenomenologically superficial, as "difficult to control" thoughts are seen in many other conditions (generalized anxiety disorder, depression, impulse control disorders, substance use disorders). A closer phenomenological overlap is the occurrence of thoughts which are ego-dystonic - in OCD, these are perceived as belonging to the self, whereas in SSD they are seen as being inserted or imposed by an external agency. Likewise, the neurobiological, neurodevelopmental, epidemiological (in terms of comorbidity) and neuropsychological links between OCD and SSD could be mentioned briefly. This would provide a clearer rationale for the current study.
3. The paper states that OCD increases the risk of subsequent SSD. This is a somewhat controversial assertion as the rate of "transition" from OCD to SSD is relatively low, whereas the occurrence of OCD / OC symptoms in SSD is somewhat more common. Recent evidence to support this statement should be presented, if available.
4. Given the distinctive nature of early-onset / childhood OCD (high comorbidity with tic disorders and externalizing disorders, poorer insight, unstable course), the decision to include both early-onset and adult-onset OCD in the same research is debatable. Was the study adequately powered to identify clear links between early-onset OCD, childhood adversity and vulnerability to psychosis? Does "collapsing" the two groups lead to a loss in precision or an underestimation of the effect size? (The researchers could justify this by pointing to difficulties in recruiting large samples of early-onset OCD or to the preliminary stage of research in this field.)
5. The current study has included patients aged 10-65. The CTQ has been shown to yield valid results in patients aged 12 and above (or 14 and above in some reports) according to different experts. Is there evidence that this tool is valid for use in children below the age of 12?
6. Details of the study's sample size / power to detect meaningful associations between specific types of trauma / neglect and psychotic vulnerability can be added to the Methods section.
7. Was family history of schizophrenia / psychosis / SSD assessed in the study participants? This is an important confounding factor as it can markedly increase psychotic vulnerability by itself.
8. Were other forms of early childhood adversity also evaluated? The CTQ mainly evaluates abuse and neglect in the familial context. However, other forms of adversity in early life (early childhood economic deprivation, bullying, exposure to traumatic stressors such as natural disasters or accidents) may also be associated with the risk of psychosis.
9. In Table 3, it would be useful to include quantitative data (i.e., correlation coefficients and significance levels) and not just colour codes. As multiple correlation analyses have been carried out, it should also be stated whether these were corrected for multiple comparisons. This is important as the strength of the reported correlations is weak to moderate (correlation coefficient 0.3 to 0.5).
10. In the Discussion, the authors should examine the possible reasons why emotional neglect, but not other forms of adversity, were associated with vulnerability to psychosis in patients with OCD. (For example: What are the neurobiological and psychological correlates of emotional neglect as distinct from other forms of abuse or physical neglect? How could these correlates be used to explain links between OCD and the risk of psychosis?)
